



# Investigating the Local Scale Influence of Sea Ice on Greenland Surface Melt

Julienne C. Stroeve[1,2], John R. Mioduszewski[3], Asa Rennermalm[4], Linette N. Boisvert[5] and Marco Tedesco[6], David Robinson[4]

[1]National Snow and Ice Data Center, Cooperative Institute for Research in Environmental Sciences, University of Colorado, 449 UCB, Boulder, CO 80309, USA.

[2]Centre for Polar Observation and Modelling, University College London, Department of Earth Sciences, Gower Street, London, WC1E6BT, UK.

[3]Center for Climatic Research, University of Wisconsin – Madison, 1225 W. Dayton St., Madison, WI 53706, USA.

[4]Department of Geography, Rutgers, The State University of New Jersey, 54 Joyce Kilmer Avenue, Piscataway NJ 08854-8045, USA.

[5]NASA Goddard Space Flight Center, Greenbelt, MD, 20771, USA.

[6]Lamont, Columbia University

## Abstract

Rapid decline in Arctic sea ice cover in the 21[st] century may have wide-reaching effects on the Arctic climate system, including the Greenland ice sheet mass balance. Here, we investigate whether local changes in sea ice around the Greenland ice sheet have had an impact on Greenland surface melt. Specifically, we investigate the relationship between sea ice concentration, the timing of melt onset and open water fraction surrounding Greenland with ice sheet surface melt using a combination of remote sensing observations, and outputs from a reanalysis model and a regional climate model for the period 1979 - 2015. Statistical analysis points to covariability between Greenland ice sheet surface melt and sea ice within Baffin Bay and Davis Strait. While some of this covariance can be explained by simultaneous influence of atmospheric circulation anomalies on both the sea ice cover and Greenland melt, within Baffin Bay we find a modest correlation between detrended melt onset over sea ice and the adjacent ice sheet melt onset. This correlation appears to be related to increased transfer of sensible and latent heat fluxes from the ocean to the atmosphere in early sea ice melt years, increasing temperatures and humidity over the ice sheet that in turn initiate ice sheet melt.



## 1. Introduction

The shrinking sea ice cover is one of the most striking features of Arctic climate change
[e.g. *Stroeve et al.*, 2012; *Serreze et al.*, 2007]. Since the late 1970s, the sea ice extent (SIE)
has declined by more than 40% in September, with smaller, yet statistically significant
negative trends in other months. These negative trends have been linked to the observed
increases in atmospheric $CO_2$, with the prospect of the Arctic Ocean becoming seasonally
ice free before the middle of this century if current emission rates continue [*Notz and*
*Stroeve*, 2016]. At the same time, the Greenland ice sheet (GrIS) has experienced increased
summer melt [e.g. *Tedesco et al.*, 2011; *Fettweis et al.*, 2011] and an increasingly negative
mass balance [*Khan et al.*, 2015]. While earlier studies found GrIS mass loss to be
balanced by ice discharge and ice melt [*van den Broeke et al.*, 2009], newer evidence
shows surface melting is now contributing 84% to the mass loss since 2009 [*Enderlin et*
*al.*, 2014]. It has further been suggested that surface melting will dominate Greenland's
contribution to sea level rise throughout the rest of this century [*Enderlin et al.*, 2014; *Fyke*
*et al.,* 2014]. Similar to the sea ice environment, an anthropogenic signal has been
identified in the observed changes of GrIS surface mass balance (SMB) [*Fyke et al.*, 2014].
While both the GrIS and sea ice environments are responding to anthropogenic
warming [*Hanna et al.*, 2008], changes in atmospheric circulation patterns that favor
increased sea ice loss and GrIS melt have also played a role. Analysis of summer (JJA) sea
level pressure (SLP) reveal statistically significant increases over Greenland and north of
the Canadian Arctic Archipelago coupled with significant negative trends over northern
Eurasia and Canada from 1979 to 2014 [*Serreze et al.*, 2016], dominated by a clear shift in
the last decade (2005 to 2014) towards large positive SLP anomalies over the central Arctic
Ocean and Greenland. This pattern favors both summer sea ice loss [e.g. *Wang et al.,* 2009;
*Ogi and Wallace*, 2007] as well as Greenland surface melt [*Hanna et al*. 2013*;*
*Mioduszewski et al.,* 2016]. Additionally, advection of warm and humid air masses appears
to be the primary factor initiating sea ice melt onset [*Boisvert and Stroeve*, 2015; *Mortin et*
*al.*, 2016]. Anomalous GrIS melting also appears to coincide with increasing water vapor
transport to the ice sheet [*Mattingly et al.*, 2016]. Thus, it is not surprising that there is a
strong inverse correlation between GrIS melt intensity (defined by *Tedesco et al.,* 2007)
and the pan-Arctic September SIE ($r$ = -0.83 from 1979 to 2015) [**Figure 1**]. Detrended
data reveal a substantially weaker inverse relationship ($r$ = -0.27), yet the year-to-year
variability between September SIE and GrIS melt remains highly correlated ($r$ = -0.69).
This would suggest that atmospheric processes fostering a high melt year also tend to foster
more summer sea ice loss and vice versa.
What about local-scale feedbacks? Changes in sea ice have strong local-scale
influences on the Arctic climate through enhanced transfer of heat and moisture between
the ocean and atmosphere, resulting in amplified Arctic warming [e.g. *Serreze et al.*, 2009;
*Screen and Simmonds*, 2010]. This is mostly manifested during the cold season, as
warming of the ocean mixed layer during summer results in increased sensible and latent
heat transfer from the ocean to the atmosphere [*Boisvert et al.*, 2015]. Other studies have
linked sea ice loss to atmospheric warming in surrounding areas during other times of the
year as well [*Comiso et al.*, 2002; *Hanna et al.,* 2004; *Bhatt et al.*, 2010, *Serreze et al.*,
2011]. Sea ice loss is additionally tied to increased tropospheric moisture, precipitation,
cloud cover, surface temperature, and decreased static stability [*Deser et al.,* 2000*; Rinke et*
*al.,* 2006*; Francis et al.,* 2009*; Serreze et al.,* 2009*; Kay et al.,* 2011*; Screen and*
*Simmonds,* 2010*; Stroeve et al.,* 2011*; Overland and Wang,* 2010*; Cassano et al.,* 2014].



Water vapor or moisture increases surface melting through its role in cloud formation and
as a greenhouse gas, results in increased downward longwave radiation and precipitation
[*Bennartz et al.*, 2013, *Doyle et al.,* 2015, *van Tricht et al.*, 2016].
This study examines whether or not local changes in the sea ice environment around
Greenland are already impacting GrIS meltwater production and therefore SMB variations.
First, we identify regions of SIC and GrIS melt covariability by applying the singular value
decomposition method. We hypothesize that regions of covariability will have consistent
trends in sea ice cover and melt production, as well as consistent trends in spring melt onset
and fall freeze up. As a second step, this hypothesis is examined with a spatial analysis of
trends for the entire study domain. Third, we investigate if a plausible mechanism for local
scale influence between SIC and GrIS is present. Specifically, we hypothesize that the
mechanism for the local scale influence is controlled by positive turbulent fluxes from the
SIC regions. Therefore, anomalous turbulent fluxes should be larger in years with early sea
ice melt onset than in later years in regions of covariability. In turn, these turbulent heat
fluxes should result in increased specific humidity and near surface temperature over the
GrIS, which should be reflected in positive net longwave radiation anomalies. Finally, a
detailed analysis, restricted to the region with evidence of local scale influence, is
performed. In this analysis, we examine the hypotheses that the timing of turbulent heat
flux anomaly perturbations over reduced sea ice areas proceeds changes in GrIS humidity
and temperature, and that wind patterns in early melt onset years are favorable for turbulent
heat flux transport from the ocean to the ice sheet. Finally, correlation and partial
correlation analysis is used to examine the influence of large scale atmospheric circulation
(here represented by the Greenland Blocking Index).

## 2. Data

### *2.1 Sea Ice and Ice Sheet Data*

Sea ice and Greenland melt extent/area calculations rely on algorithms applied to
satellite passive microwave data from the Nimbus-7 Scanning Multichannel Microwave
Radiometer (SMMR: 1978-1987) and the DMSP Special Sensor Microwave/Imagers
(SSM/I and SSMIS: 1987-present). Specifically, we use several sea ice metrics derived
from the NASA Team SIC algorithm [*Cavalieri et al.*, 1996, updated 2008] and distributed
by the National Snow and Ice Data Center (NSIDC). The data set spans October 1978 to
present, providing daily (or every other day during the SMMR era) SIC estimates. Using
the SIC, we additionally calculate the open water fraction (OWF) as well as the length of
the ice-free season, defined as the number of days each year with ice concentration less
than 15% [see *Parkinson*, 2014].
Changes in the timing of melt onset (MO) and freeze-up (FO), in addition to total melt
season length over sea ice, are computed following *Markus et al.* [2009]. This study uses
an updated version of the algorithm that bias corrects for intersensor calibration issues
found between the F17 and F13 sensor and evaluates early melt onset (EMO),
corresponding to the first day of MO, the continuous MO and the continuous FO.
GrIS melt extent is an estimate of the daily spatial extent of wet snow using the *Mote et*
*al.* [2014] algorithm and distributed by NSIDC. From the binary melt/no melt
classification, GrIS MO and FO dates were calculated for each pixel and each year from
1979 to 2015. We defined the start of the MO and FO as the first occurrence of a 5-day
continuous melt or freeze-up period. Melt duration was calculated as the number of days





between MO and FO. EMO was also determined and defined as the first time a spurious
melt event lasting at least one day was recorded.
Besides mapping the GrIS melt extent and timing of MO and FO, we use meltwater
production and 850 hPa wind as simulated by Modèle Atmosphérique Régional (MAR)
v3.2 regional climate model [*Tedesco et al.,* 2013]. MAR is a three-dimensional coupled
atmosphere-land surface model that uses reanalysis data at its lateral boundaries. In this
study, MAR is forced with data from ERA-40 for the period 1979–2002 and ERA-Interim
for the period 2002–2015 and outputs are produced on a polar stereographic projection
with an approximate grid cell size of 25 x 25 km to match the passive microwave-derived
fields. MAR's atmospheric model is coupled to the 1-D Surface Vegetation Atmosphere
Transfer scheme, SISVAT [*Gallée and Schayes*, 1994; *De Ridder and Gallée*, 1998], which
simulates surface properties and the exchange of mass and energy. SISVAT incorporates a
snow model based on the CROCUS snowpack model [*Brun et al.*, 1992]. MAR has been
validated through comparison with ground measurements [e.g. *Lefebre et al.,* 2003; *Gallée*
*et al.,* 2005; *Lefebre et al.,* 2005], satellite data [e.g. *Fettweis et al.,* 2005, 2011; *Tedesco et*
*al.,* 2011, *Alexander et al., 2014*], and applied to simulate long-term changes in the GrIS
SMB and surface melt extent [*Fettweis et al.,* 2005, 2011; *Tedesco et al.,* 2008, 2011;
*Tedesco and Fettweis, 2012*]. Data are freely available from an online repository [*Tedesco*
*et al.,* 2015].
Meltwater production was used for grid cells classified by MAR as greater than 99%
ice sheet to mask the tundra region of Greenland. In addition, meltwater production values
of less than 1 mm day$^{-1}$ in all grid cells were recoded to zero to account for MAR's scaled
output. This threshold could be considered a conservative approximation of the occurrence
of surface melt [*Fettweis et al., 2011, Figure 2*]. Finally, grid cells were masked in the
interior ice sheet where mean monthly meltwater production does not exceed 1 mm day$^{-1}$ to
account for spurious correlations arising from a very limited number of dates that result in
nonzero mean monthly values of meltwater production.
Trends for each pixel (or regional averages) are only computed if at least 30 years of valid
data are found at that pixel. This ensures statistics are not biased by changes in spatial extent of
the sea ice or Greenland melt. However, Greenland melt has been observed to extend to higher
elevations in recent years, and in 2012 nearly the entire ice sheet experienced melt events [e.g.
*Nghiem et al*., 2012]. Regional means are area-weighted. Trends are computed using linear-
least squares and statistical significance is evaluated with a student T-test.

## 2.2 Atmospheric Data

Geopotential heights at 500 hPa and hourly 10 m wind speeds were obtained from
NASA's Modern Era Retrospective-Analysis for Research and Applications (MERRA)
products [*Bosilovich et al.*, 2011; *Cullather and Bosilovich*, 2011a, 2011b; *Rienecker et al.*,
2011]. MERRA is run on a 1/2° latitude by 2/3° longitude global grid with 72 hybrid-sigma
vertical levels to produce analyses from 1979 to present. MERRA has been evaluated
extensively since its release [*Cullather and Bosilovich*, 2011b; *Kennedy et al.*, 2011;
*Reichle et al.*, 2011] and has compared favorably with other reanalysis products in the
Arctic [*Zib et al.*, 2012; *Cullather and Bosilovich*, 2011; *Lindsay et al.*, 2014].
We also utilize atmospheric variables from NASA's Atmospheric Infrared Sounder
(AIRS), designed specifically to map atmospheric water vapor content. This instrument has
been used in several recent studies to document atmospheric changes and impacts on sea
ice in the Arctic [e.g. *Boisvert and Stroeve*, 2015; *Stroeve et al.*, 2014; *Serreze et al.*, 2016].
While the data record is rather short (begins in September 2002), it provides twice daily





global coverage at 1-degree spatial resolution of several key atmospheric variables,
including skin and air temperature, precipitable water, cloud fraction and specific humidity.
In this study we utilize the Level 3 Version 6 skin temperatures, 1000 hPa air temperature,
effective cloud fraction, near surface specific humidity and total precipitable water.
Additional variables derived from AIRS data products include the moisture flux [*Boisvert*
*et al.*, 2013; 2015], turbulent sensible heat flux and downwelling longwave radiation
[*Boisvert et al.,* 2016].

### 3. Methods

### 3.1. Region of Interest and Study periods

For local assessment of sea ice changes and corresponding ice sheet changes, we define 5
sea ice and 5 adjacent ice sheet regions. Since we are examining the potential influence of the
ocean on the ice sheet, it makes sense for the ocean regions selected to define the ice sheet
boundaries, rather than the other way around. The definition of the sea ice boundaries comes
from the International Hydrographic Organization, and we define 5 sea ice regions: Baffin Bay,
David Strait, Lincoln Sea, Greenland Sea and the North Atlantic together with associated
Greenland regions [**Figure 2**]. For the ice sheet, each region is defined along a topographical
divide. While there are many local topographical divides, only those regions that matched the
ocean delineations were selected.
We use two study periods. First, we do analysis from 1979 to 2015 when analyzing sea ice,
melt extent and MAR model outputs. Second, AIRS data analysis is applied from 2003 to 2015
since a full year of data collection didn't begin until 2003.

### 3.2 Relationship between SIC and GrIS melt

To investigate covariability between summer SIC, GrIS melt water production, and 500
hPa geopotential heights, singular value decomposition (SVD) was applied to two fields at
a time to produce pairs of coupled spatial patterns that explain their maximum mean
squared temporal covariance [*Bretherton et al.*, 1992].
The temporal evolution of each pair's corresponding pattern in the two datasets is
represented by the pair's associated expansion coefficients (EC), where subscripts GrIS,
SIC and 500 denote the EC for ice sheet melt, sea ice concentration, and 500 hPa heights,
respectively. These ECs were used to calculate heterogeneous correlation (HC) maps,
which show the correlation coefficients between each EC and the opposing data field. SVD
has widely been used to investigate coupled modes of variability, including relationships
between Arctic sea ice and snow cover [*Ghatak et al.*, 2010], and Arctic sea ice and
atmospheric variables [*Stroeve et al.*, 2008].
To further investigate how SIC in these regions is related to GrIS melt, SIC for both
regions was spatially aggregated, de-trended and correlated with de-trended time series of
GrIS meltwater production and the Greenland Blocking Index (GBI), respectively [*NOAA*,
2015]. The GBI is defined as the 500 hPa geopotential height field averaged between $20° -$
$80°$ W, $60° - 80°$ N [*Fang*, 2004; *Hanna et al.*, 2013], and is used as a metric for large-
scale atmospheric circulation patterns over Greenland. To remove the influence of the GBI
on both SIC and GrIS melt, we performed a partial correlation analysis of SIC in each
region and GrIS meltwater production after the trends in GBI were removed [e.g. *Cohen et*
*al.*, 2003].



### 3.3 Energy Balance

Following *Koenig et al.* [2014], the net heat flux into the atmosphere ($F_{net}$) emitted
from the ocean is defined by:
(1)

$$F_{net} = Q_h + Q_e + LW - SW$$

where SW is the downward shortwave radiative flux at the surface, LW is the net upward
longwave radiation, $Q_h$ is the sensible heat flux, or heat transferred from the surface to the
atmosphere by turbulent motion and dry convection, and $Q_e$ is the latent heat flux, or heat
extracted from the surface by evaporation. If the sum of the four right-hand side terms is
positive, there is a net flow of heat from the surface to the atmosphere and vice versa.
Previous studies have looked at the strong seasonality in $F_{net}$ over the Arctic Ocean
[e.g. *Serreze et al.*, 2007], with strong downward fluxes in summer and large upward fluxes
in January associated with heat gain and loss, respectively, in the subsurface column.
Updated trends from NCEP/NCAR reanalysis confirm that $F_{net}$ trends are small in winter
(January to April), except in the Barents Sea as a result of reduced sea ice and increased
oceanic heat flux [*Ornaheim et al.*, 2016] and also within Baffin Bay, again a result of less
winter ice cover. Thus, in these two regions there is a transfer of heat from the ocean to the
atmosphere during the winter months, which may spread over the sea ice areas and limit
winter ice growth. In summer however (May to August), the direction is generally reversed
with large heat fluxes from the atmosphere going towards the surface.
In this study we focus on how early sea ice retreat, as indicated by early melt onset
during the transition from winter to summer, impacts the heat and moisture fluxes over
early formed open water areas, and whether or not this is sufficient to impact Greenland
melt. Towards this end, we composite the turbulent fluxes in Eq. 2 for low and high sea ice
years, specific to each individual region analyzed using the AIRS data, with positive fluxes
showing energy transfer from the surface to the atmosphere. We use the criteria of
anomalies in melt onset exceeding 1 standard deviation for each region when compositing.
All data are detrended by subtracting the linear trend before computing the composites.

## 4. Results

We begin with an assessment of the large-scale relationship between SIC and
Greenland melt and its spatial covariability (4.1). This is followed by an analysis of
changes in the sea ice cover surrounding Greenland, both in terms of SIC and OWF (4.2),
followed by analysis of the timing of sea ice MO onset and FO, and its relationship with
Greenland MO (4.3). Finally, turbulent heat and moisture flux changes composited for
early and late melt onset years are examined (4.4) and large-scale influences are examined
in section 4.5.

### 4.1 Relationship between Sea Ice and Greenland Melt

The leading SVD mode explains the majority of the mean spatial covariance between
monthly GrIS meltwater production and SIC in June and July (62%, 73%, respectively) and
less than half (42%) in August. HC maps reveal opposing sign of the correlations between
the map pairs [**Figure 3**: columns 1 and 2; and columns 3 and 4] indicating an
anticorrelation, meaning that increased ice sheet melt extent covaries with decreased sea
ice area (it is irrelevant in the HC maps which is positive and which is negative).
Specifically, the covariability of GrIS meltwater production and SIC, expressed as





correlations on an HC map, show that sea ice and ice sheet melt strongly covary in two
general regions, namely Baffin Bay/Davis Strait in June, and a large part of Beaufort Sea in
June and July [**Figure 3(a), (e) and (i)**]. In June, SIC in both the Baffin Bay/Davis Strait
and the Beaufort Sea regions have strong correlations with $EC_{GrIS}$, $|r| > 0.70$), and GrIS
meltwater production is highly correlated with $EC_{SIC}$ for the majority of the unmasked ice
sheet surface [**Figure 3b**]. The strong correlation in the Beaufort Sea persists in July but
not in Baffin Bay/Davis Strait, and neither exhibits a significant correlation in August
[**Figure 3(e) and (i)**]. At the same time, GrIS meltwater production correlations with $EC_{SIC}$
are less expansive over the ice sheet in July and August, particularly in southern Greenland
[**Figure 3(f) and (j)**].
In the second SVD analysis of 500 hPa geopotential heights and GrIS melt water
production, the leading SVD mode explains the majority of mean spatial covariance of the
two variables in June and July (79% and 60%, respectively), but less than half in August
(37%), which are similar values to the leading SVD mode for GrIS melt and SIC [**Figure
3(c), (g) and (k)**]. The HC maps show a strong tendency for positive height anomalies
centered on the Greenland side of the Arctic, though this area shrinks in July and August
[**Figure 3(c), (g) and (k)**]. As before, this spatial pattern covaries with GrIS melt water
production over most of the ice sheet in June, but is somewhat more restricted in extent in
July and August. While SIC and GrIS melt extent covary regionally, large parts of the same
areas of the GrIS melt extent region also covary with 500 hPa geopotential height fields.
The similar spatial patterns in GrIS melt covariability with SIC and 500 hPA geopotential
height fields suggest that the large-scale circulation may be a dominant explanation for the
SIC – GrIS melt covariabilty. Before this possibility is examined more closely, we analyze
trends in SIC and GrIS melt patterns and timing.

### 4.2 Changes in the Sea Ice Cover around Greenland

The above analysis suggests a local-scale influence from SIC on GrIS melt within
Baffin Bay and Davis Strait during June. This region of high SIC-GrIS covariability has
experienced a sharp drop in SIC since 1979 [**Figure 4**]. In Baffin Bay and Davis Strait, SIC
trends are negative in all seasons, and are particularly large in winter (DJF), spring (MAM)
and summer (JJA) [**Figure 4a-d**]. In contrast, SIC trends in the East Greenland Sea are
mixed, which may in part explain the lack of covariability within this region. Adjacent to
the Greenland's east coast, positive SIC trends occur throughout winter and spring. Further
east, reductions in SIC are confined to the area where the Odden used to form (c.f. Figure
2). During summer and fall, negative SIC anomalies persist along eastern GrIS, though
they remain smaller than on the western side. North in the Lincoln Sea region, there is
essentially no change in SIC year-round except for slight negative trends in summer.
Negative SIC trends have resulted in longer open water periods surrounding Greenland
[**Figure 4e**]. Trends in annual open water days are mostly positive everywhere, the
exceptions being the Lincoln Sea, which remains ice-covered year around, and the southern
part of Davis Strait towards the Labrador Sea, a region where ice has expanded during
recent winters. In some locations within Baffin Bay and the East Greenland Sea the number
of open water days has increased by as much as 30 to 40 days per decade, but regionally
averaged trends are generally on the order of 2 weeks per decade.
The strength of the OWF trends and exact timing of when these trends peak around the
GrIS reveal large spatial differences [**Figure 5**]. The largest OWF trends occur in Baffin
Bay during week 26 (third week of June), and are on the order of 10% dec$^{-1}$, with a
secondary peak during week 44 (end of October). Further south in Davis Strait, OWF are





positive throughout winter and into July (~5% dec$^{-1}$), reflecting both earlier ice retreat and
later winter ice formation, with the largest trends during week 52 (6% dec$^{-1}$). East of
Greenland, positive OWF trends are found throughout the year in the Greenland Sea, but
are considerably weaker than found in Baffin Bay and Davis Strait. Finally, Lincoln Sea
OWF trends are mostly negative (except in June and August), though trends are generally
less than 1% dec$^{-1}$, and are not statistically significant. For comparison the Arctic Ocean
OWF trends are also shown, showing peak OWF trends around week 38 (mid-September),
reflecting the timing of the pan-Arctic sea ice minimum.

### 4.3 Changes in the Melt Season

We next examine if there is a link between the timing of EMO, MO, and FO over sea
ice and over GrIS. The link between MO and the timing of ice retreat has already been
established, with correlations between the detrended melt onset and detrended ice retreat
dates greater than 0.4 [See Figure S10, *Stroeve et al., 2016*].
Climatological regional mean values of EMO, MO, FO show that melt begins earlier
and freeze-up happens later over the sea ice than it does on the ice sheet, and can be largely
explained by temperature dependencies on elevation [**Table 1**]. In western Greenland, the
continuous MO period for sea ice begins about 9 days earlier than on the ice sheet in the
Baffin Bay region, and 15 days earlier in the Davis Strait region, whereas ice sheet FO
occurs on average in early to mid-September, compared to the end of October (Baffin Bay)
to the end of November (Davis Strait) over the adjacent sea ice. Similarly, in the Greenland
Sea region, MO begins around 20 days earlier over the sea ice than on the ice sheet and FO
happens about a month later. In contrast, the Lincoln Sea region exhibits similar timing in
both MO and FO, which may be explained by the fact that this is the smallest region, and
also the region furthest north where most melting will only occur at lowest GrIS elevations.
Since there is little sea ice in the North Atlantic (e.g. regionally the open water season lasts
for 360 days), MO and FO dates are not meaningful, but generally show values similar to
as that observed in Davis Strait.
EMO, MO and FO trends for SIC and GrIS are of the same sign, indicating an overall
lengthening of the melt season over the last 37 years in both environments [**Figure 6**].
Baffin Bay experiences the largest trends towards earlier MO and later FO, with regionally
averaged trends of -8.3 and +7.8 days dec$^{-1}$, respectively, statistically significant at 99%
confidence [**Table 2**]. This has led to an increase in the melt season length on the order of
16 days per decade. GrIS trends in the same region are typically smaller, especially in
regards to the timing of freeze-up (4.6 days dec$^{-1}$) and melt season duration (11.1 days dec$^{-1}$
). In contrast, larger statistically significant trends in both MO and FO are seen over the
Davis Strait GrIS region, leading to a lengthening of the melt season that is larger than over
the adjacent sea ice (18.7 days dec$^{-1}$ compared 11.7 days dec$^{-1}$).
On Greenland's eastern side, similar ice sheet/sea ice MO trends are observed, but sea
ice FO trends are smaller, and not statistically significant. The exception is the North
Atlantic region, which exhibits large positive FO trends of 8.9 days dec$^{-1}$, resulting in an
overall increase in melt season duration of 16.3 days dec$^{-1}$. However, given the low
frequency of sea ice in this region, caution is warranted when interpreting these trends
since ocean dynamics play a large role in the year-to-year variability in these values.
Nevertheless, the largest trends in melt season duration over the eastern GrIS are also
found in the North Atlantic sector (22.1 days dec$^{-1}$), primarily a result of earlier MO. The
Greenland Sea GrIS sector also exhibits large trends in melt duration (14.4 days dec$^{-1}$), but
earlier MO and later FO play a nearly equal role here. Interestingly, the Lincoln Sea GrIS



region also displays large trends in melt season duration (12.7 days dec$^{-1}$), considerably
larger than seen over the adjacent sea ice (5.5 days dec$^{-1}$). While the climatological mean
timing of MO and FO is broadly similar over both the sea ice and the GrIS in the Lincoln
Sea GrIS region, there has been a trend towards much later freeze-up (6.8 days dec$^{-1}$).
Finally, we examine whether there is synchronicity in the timing of melt onset and
freeze-up between the sea ice and the ice sheet. In the Baffin Bay sector, the correlations
between the sea ice and ice sheet MO and FO (respectively) exceed *0.6; p=0.001*. High
correlations (*r>0.6*) are also seen in the Lincoln Sea sector and for EMO in the Greenland
Sea sector (*r=0.6; p=0.001*). Correlations are reduced when MO, FO and EMO records are
detrended, yet remain significant in the Baffin Bay and Lincoln Sea regions: detrended
correlations for sea ice and the ice sheet EMO, FO and melt season duration exceed *r=0.5,*
*p=0.001* in Baffin Bay as well as the Lincoln Sea in regards to the MO, *p=0.002*.
Elsewhere, no significant relationship is found.

### 4.4 Impact of sea ice changes on surface energy fluxes

Next we examine the relationship between early and late MO and variations in
atmospheric moisture and heat fluxes using lag-correlation and composites for early and
late MO years. We begin with an assessment of the differences in the strength of turbulent
fluxes between early and late MO years. All months are shown to allow for both an
assessment of what drives early MO over sea ice as well as to determine how early sea ice
MO influences the overlying atmosphere [**Figure 7**].
On average, the transfer of latent heat flux occurs from the ocean to the atmosphere
year-round in all regions, except the Lincoln Sea in Sep-May, and Baffin Bay in Dec-Feb.
In Baffin Bay and Lincoln Sea, latent heat flux transferred to the atmosphere is small until
the sea ice begins to break up and melt in the summer and moisture is released from the
previously ice-covered ocean. Latent heat fluxes are directed into the atmosphere year-
round in Davis Strait and Greenland Sea due to large areas of ice-free ocean that persists
throughout the year.
Sensible heat flux is generally directed towards the surface for regions that are 100%
sea ice covered during the cold season months (e.g. Baffin Bay and the Lincoln Sea) and
then switches towards the atmosphere as the sea ice retreats in summer (Baffin Bay only).
Regions that have large fractions of open water year-round generally have a net sensible
heat flux transfer towards the atmosphere year-round, though some exceptions occur.
Greenland Sea and Davis Strait exhibit sensible heat flux to the atmosphere in early spring
and late fall (October-December) when the ice-free ocean surface is much warmer than the
overlying air; due to the higher heat capacity of water, the opposite is true for ice-covered
regions.
A larger amount of sensible and latent heat flux tends to enter the atmosphere in the
spring during early MO years in all regions. However, the Baffin Bay region is the only
region with a majority of positive fluxes throughout the year. When melt happens early in
Baffin Bay, the additional sensible and latent heat fluxes result in ~14 Wm$^{-2}$ entering the
atmosphere in spring (March-June) and ~25 Wm$^{-2}$ in autumn (September-December) due
to a later FO. In contrast to Baffin Bay, turbulent flux anomalies in early MO years from
Davis Strait and Lincoln Sea show no strong consistent pattern and switch between positive
anomalies throughout the year. Compared to Baffin Bay, Davis Strait, which is further
south, has larger latent heat fluxes entering the atmosphere between February-August
during years with earlier MO, whereas sensible heat flux into the atmosphere is only larger
during early MO years in February, April and November, reflecting both early MO (April)



and later FO (November). Over the Lincoln Sea there are no fluxes of heat or moisture into
the atmosphere during the late fall, winter and early spring due to the solid sea ice pack.
However, by June there is an additional ~12 W m$^{-2}$ of turbulent flux energy transferred to
the atmosphere during early melt years. This generates smaller turbulent fluxes in July due
to warmer air temperatures than when melting has just begun in late MO years. The early
MO year turbulent flux anomalies from Greenland Sea are different from the other three
regions, as there is more heat and moisture entering the atmosphere in January, March,
October and December during early MO years.

Sensible and latent heat fluxes transfer heat and moisture into the local atmosphere and
can cause the temperature and humidity to increase, which in turn should produce larger
downwelling longwave flux at the surface due to the greenhouse feedback effect. Thus one
would expect to see a larger net longwave flux (downwelling – upwelling) at the surface
during early MO years when the local atmosphere contains more heat and moisture. We see
evidence of this occurring until roughly July as there is more net longwave directed
towards the surface of the ice sheet in most regions when the sea ice melts earlier **[Figure
8]**. In August the surface net longwave flux turns largely negative during early MO years,
partly because the warmer ice sheet results in dominance of upwelling radiation fluxes, and
partly because there is less of an influence of early season conditions.

The increase in heat and moisture into the atmosphere from the surrounding ocean in
early MO versus late MO years and subsequent increase in energy at the ice sheet surface is
shown in more detail for Baffin Bay in **Figures 9(a)** and **(b).** In April and May (day 1 to 61
in Figure 9), there appears to be an out-of-phase relationship between latent heat flux over
Baffin Bay and the specific humidity over the adjacent ice sheet, with pulses of moisture
coming from the ocean surface being followed about a week later with rising specific
humidity over the ice sheet. A similar pattern is observed between ocean sensible heat flux
and near surface air temperature over GrIS. In June and July (day 61 to 92), latent and
sensible heat flux anomalies for early/late MO years fluctuate around zero, which suggests
these fluxes are similar between early and late MO years. In contrast, the specific humidity
and temperature are higher in late MO years over the ice sheet in July (negative anomalies
in Figure 9a and 9b). This could be due to a roughly one-month delay in late MO years
compared to early MO for the sea ice, which causes increases in the temperatures and
humidity later in the season (July) over the ice sheet. From the timing of early sea ice MO
(dotted blue line) to early GrIS MO (dotted blue, highlighted red line), large fluxes of
moisture and heat released via the latent and sensible heat flux from the ice/ocean surface
precede elevated humidity and temperature over the ice sheet.

One-week running lagged correlations between latent heat flux from the ocean and
specific humidity over the ice sheet show large positive correlations during early MO years
[Figure 9a top, solid blue lines], suggesting increased evaporation from earlier MO over
sea ice may be driving the observed increase in specific humidity over the ice sheet one
week later. A one-week lag was chosen because sea ice and GrIS MO in Baffin Bay occur
about 9 days apart on average, and also because water vapor in the troposphere has a
residence time of about two weeks. These three highly correlated events precondition the
ice sheet for earlier MO by increasing the specific humidity and thus the downwelling
longwave flux earlier in the spring. In late MO years, the sea ice/ocean does also appear to
play a small role in initiating MO on the ice sheet. Large amounts of latent heat are
released from the surface in Baffin Bay at the timing of late MO, which in turn is correlated
to increases in specific humidity over the ice sheet directly before MO, initiating melt
(solid green lines). Since Baffin Bay MO is much later (~1 month) in late melt years,



excess moisture into the atmosphere is delayed. Though because the environment is already
warming seasonally, it does not require extra preconditioning for the melt to begin on GrIS
compared to early melt years. This case is very similar to sensible heat flux released from
Baffin Bay and ensuing temperature over the ice sheet [**Figure 9b**]. Comparing these 1-
week lagged correlations to a zero-lag correlation (not shown), correlations for all variables
in early and late MO years are highly negative, meaning they are out of phase.
Note also there are instances in April when both early and late melt years exhibit high
correlations between either sensible or latent heat from the sea ice region and specific
humidity or temperature over GrIS one week later. This may be related to opening of the
North Water Polynya [*Boisvert et al.*, 2012]. As the open ocean is relatively warm
compared to the overlying air in April, heat and moisture fluxes enter the atmosphere and
are subsequently transferred over the ice sheet, increasing the specific humidity and air
temperature.
In summary, sea ice in Baffin Bay/Davis Strait and the adjacent ice sheet surface
conditions appear connected. MO and breakup of the sea ice triggers enhanced flux of heat
and moisture into the atmosphere, which are observed over the ice sheet within a week.
This results in a warming and moistening the local environment and preconditions the ice
sheet for melt in early MO years. Therefore, when the MO of the sea ice is earlier, MO of
GrIS is earlier and vice versa.

### 468 *4.5 Influence of large scale atmospheric variability on Baffin Bay*
### 469 *and Beaufort Sea*

The SVD analysis (4.1) indicated that both Baffin Bay/Davis Strait and the Beaufort
Sea are regions with SIC and GrIS melt water production covariability. In the case of
Baffin Bay/Davis Strait, this was supported by the melt and turbulent heat flux analysis.
Next we examine the influence of the large-scale atmospheric variability on this
covariability using Pearson correlation and partial correlation.
In the Beaufort Sea, both 500 hPa heights and SIC closely covary, particularly in June
[**Figure 10a**], in concert with high SIC covariance in this region with EC$_{GrIS}$ in the HC
maps [**Figure 3**]. Here, the positive correlations between SIC and GrIS melt weaken
significantly after June with almost no correlation by August [**Table 3**]. The strong
relationship between Beaufort SIC and GrIS melt in June is reduced considerably when the
GBI index is removed via partial correlation, as significant correlations remain only in
southeast Greenland.
The correlation between SIC in Baffin Bay/Davis Strait and geopotential heights is
relatively strong but not as extensive in June, while this signal mostly disappears in July
and especially August [**Figure 10d; Table 3**]. This is associated with a weakening Baffin
Bay SIC correlation with EC$_{GrIS}$ in the HC maps [**Figure 3(a), (e) and (i)**]. Statistically
significant correlations with meltwater production are focused on the west side of the ice
sheet in June [**Figure 10e**], but are minimal in July and August when correlations over only
7% and 2% of the respective unmasked ice sheet area are statistically significant. Partial
correlation analysis indicates that the GBI explains approximately two thirds of this
correlation in each month, though this still leaves the possibility that variations in Baffin
Bay sea ice are in part responsible for the correlation with surface melt in western
Greenland.
Because there is a potential local influence from Baffin Bay and Davis Strait, we next
focus on GrIS melt only in west-central Greenland. The highest and lowest melt years in



west-central Greenland (after removing trends) consistently correspond to patterns of
anomalous SIC and geopotential heights in these years [**Figure 11**]. These variables show
much less variation by month, though a weaker relationship appears particularly in the
height field, which follows results from the SVD analysis [**Figure 11 (g)-(l)**]. Additionally,
a strong SIC pattern is evident not just in western Greenland but consistently on the east
side of Greenland that is equally as strong [**Figure 11(a)-(f)**]. This suggests that the
processes responsible for this signal expression to the west of Greenland probably also
exist on a large enough scale to have an effect of similar strength on sea ice off
Greenland's east coast; most likely a persistent ridge or trough, as suggested by the above
results. By August, sea ice in Baffin Bay has melted in most years, but positive anomalies
in SIC still appear in the lowest Greenland melt years [**Figure 11(f)**].
In summary, the SVD analysis suggest covariablity between SIC and GrIS melt in the
Baffin Bay region (Fig. 3) that cannot fully be explained by large scale atmospheric
patterns (Fig 10 and 11). Examination of a set of hypotheses applied for the entire GrIS and
surrounding seas shows that trends and patterns in the Baffin and Davis Strait regions are
consistent with local scale influence [**Table 4**]. In contrast, no other regions have evidence
of covariability or trends and patterns consistent with local scale influence.

## 5. Discussion

Sea ice and Greenland ice sheet melt demonstrate significant covariability during the
summer, particularly June. While the majority of this relationship appears related to
simultaneous atmospheric circulation forcing, analysis over Baffin Bay/Davis Strait and the
adjacent ice sheet indicates that the covariability may additionally include a local-scale
influence. This is in agreement with previous work by *Rennermalm et al*. [2009] who found
the SIE and GrIS surface melt extent to co-vary in the western part of the ice sheet, though
the strongest relationships were found in August rather than June. Part of the discrepancy
might be explained by the study period. This study extends through 2015 and includes
years with larger anomalies in both SIC and GrIS melt. However, June is the time of year
with the largest trends in OWF, reflecting earlier development of open water at a time
when the atmosphere is still relatively cold. Thus, it is not surprising that we find stronger
covariability in June and a link with melt onset. An additional area of covariability in terms
of melt onset timing is also seen in the Lincoln Sea sector.
While statistical analysis suggests a local-scale influence may be present on the western
side of the ice sheet, the ability for the sea ice to influence GrIS melt depends on having
anomalous heat and moisture sources that can travel to the ice sheet. In this study we find
that turbulent fluxes are often larger during early MO years in the spring and fall because
areas where the ocean is ice-free tends to be warmer than that of the air, due to the higher
heat capacity of water. Both latent and sensible heat fluxes are larger and more positive
(from the ocean surface to the atmosphere) during early MO years, resulting in increased
air temperature and specific humidity especially in May when the atmosphere is ~2 K
warmer and ~0.5 g kg$^{-1}$ wetter. This excess heat and humidity increases downwelling
fluxes to the ice sheet earlier in the year, preconditioning the ice sheet and triggering melt
(also shown in Figure 8). For late MO years, this phenomena occurs later in the season, and
this is most likely why we see larger fluxes during late MO years in the summer months
(i.e. July depending on the climatology of the region). This is specifically true for Baffin
Bay, where throughout the winter months the region is completely covered by sea ice,
creating a barrier between ocean-atmosphere energy exchanges. This is also valid for the



Lincoln Sea in the content of melt ponds and a higher occurrence of leads forming on the
thick multi-year ice during the summer months.
Turbulent fluxes from increased open water can reach well above the boundary layer
[e.g. *Yulaeva et al.*, 2001], but this depends on the frequency of spring and early summer
inversions that cap the atmospheric boundary layer. Furthermore, if katabatic winds are
persistent at the ice edge, this will keep onshore flow from reaching the ice sheet [*Noël et*
*al.,* 2014], though a possibility remains for mixing in the boundary layer via a barrier wind
mechanism [*van den Broeke and Gallée*, 1996]. Analysis of daily winds around the timing
of sea ice melt, show that during early MO years over the sea ice, wind direction is from
the open water areas of Baffin Bay onto the GrIS, which helps support our claims that
earlier melt onset in part drives early melt over Greenland [**Figure 12**]. In late MO years,
the wind direction is reversed.
Finally, we note that SVD analysis reveals the strongest relationship between GrIS melt
and sea ice variability occurs within the Beaufort Sea. This appears to be related to the
positioning of a ridge near Greenland that enhances both ice sheet melt and sea ice retreat
as stronger easterlies help to circulate ice west out of the Beaufort Sea. SVD analysis
shows the covariability in June is reduced considerably when the GBI index is removed via
partial correlation, evidenced by the large reduction in percentage of grid cells with a
significant correlation (not shown). This mechanism has been identified previously as a
way to transport and melt ice between the Beaufort Sea and the East Siberian Sea [*Rogers*,
1978; *Maslanik et al.*, 1999]. We speculate that no mechanism originating from sea ice
variability directly influences GrIS melt from a distance of hundreds of kilometers away,
though Liu et al. (2016) argue that sea ice loss within the central Arctic has favored
stronger and more frequent blocking events over Greenland.
In 2012, as the sea ice cover reached its all-time record low September extent, the
Greenland ice sheet also experienced a record amount of surface melt and ice mass loss
[*Tedesco et al.,* 2013]. Several explanations have been put forth to explain this anomalous
melt, including increased downwelling longwave radiation from low-level liquid clouds
[*Bennartz et al.*, 2013], advection of moist warm air over Greenland [*Neff et al.*, 2014] and
dominance of non-radiative fluxes [*Fausto et* al., 2016]. While this event was likely a result
of atmospheric circulation patterns that transported warm, humid air over the southern and
western part of the ice sheet, the sea ice melt season began a week earlier than the 1981-
2010 long-term mean over Davis Strait and 3 days earlier over Baffin Bay. This earlier
melt onset of the sea ice may have provided an additional source of warm, moist air over
the adjacent ice sheet.

## 6. Conclusions

Based on multiple lines of statistical evidence, we identified western Greenland as a
region where direct influence from sea ice on the GrIS SMB is possible. SVD analysis
revealed that extreme melt years over the adjacent ice sheet are accompanied by strong SIC
anomalies within Baffin Bay and Davis Strait that would be expected if a local-scale
thermodynamic influence were occurring. This is true even after near surface temperature
and climate index influences are removed.
The covariance is strongest in June, which may be partially due to the lower variability
in interannual June meltwater production over the entire ice sheet relative to the rest of
summer, with a standard deviation simulated by MAR of 0.84 mm water equivalent day$^{-1}$
compared to 0.95 in August and 1.12 in July. Additionally, June variability in sea ice may



have a greater potential to influence GrIS melt given that the ice sheet is transitioning into its warm season regime and reaching the freezing point for the first time in many locations. This is further confirmed through correlations between the timing of melt onset, which occurs on average 9 days earlier over the sea ice than on the adjacent ice sheet, and in turn allows for earlier development of open water and enhanced transfer of turbulent heat fluxes from the ocean to the atmosphere. More heat and moisture is transported to the local atmosphere from the ice-free ocean surface via turbulent fluxes in years when sea ice melts earlier. Daily wind field analysis suggests these enhanced turbulent fluxes are transferred to the ice sheet, allowing the local atmosphere over the GrIS to warm and become more humid, which in turn impacts the net downwelling longwave flux, helping precondition the surface for earlier melt onset.

However, despite evidence of a possible local-scale influence, all analysis incorporating 500 hPa height anomalies suggests that the large-scale atmospheric circulation remains the primary melt driver in this part of the ice sheet as well as for the ice sheet as a whole. Anomalous atmospheric circulation features include increased frequency of the negative phase of the Arctic Dipole [*Overland and Wang*, 2010] and a persistently negative summer North Atlantic Oscillation [*van Angelen et al.*, 2013]. Continued Arctic amplification and associated shifts in Arctic atmospheric circulation and their persistence will theoretically continue to enhance warming in the vicinity of Greenland [*Francis and Vavrus*, 2012, 2015]. Nevertheless, our study suggests a local response is also possible, and as the sea ice cover continues to retreat around the Greenland ice sheet, this should present further opportunities for local enhancement of summer ice sheet melt.

**Acknowledgements**

This work was funded by the National Science Foundation PLR 1304807. All data used in this study were obtained from free and open data repositories. Detailed information is provided in the methods section. The work of Linette Boisvert was funded from NASA ROSES 2012 IDS proposal: 12-IDS12-0120. AIRS data are freely available at www.airs.jpl.nasa.gov and MERRA2 data can be found at gmao.gsfc.nasa.gov.

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

00147.1.






**Table 1.** Climatological (1981-2010) mean values in length of open water season, together
with climatological dates in early melt onset (EMO), continuous melt onset (MO),
continuous freeze-up (FO) and melt season duration for 5 sea ice regions (excluding the
North Atlantic where little sea ice exists). Corresponding mean dates in melt onset, freeze-
up and duration are also shown for the Greenland drainage basins.

| Region | Length of Open Water Season (days) | EMO (day of year) | MO (day of year) | FO (day of year) | Melt Duration (days) |
|---|---|---|---|---|---|
| *Sea Ice Regions* | | | | | |
| Baffin Bay | 104 | 146 | 155 | 291 | 136 |
| Davis Strait | 220 | 133 | 143 | 321 | 188 |
| North Atlantic | 360 | 110 | 134 | 313 | 178 |
| Greenland Sea | 227 | 143 | 148 | 267 | 119 |
| Lincoln Sea | 0 | 162 | 172 | 232 | 60 |
| *Greenland Ice Sheet Drainage Regions* | | | | | |
| Baffin Bay | | 162 | 164 | 232 | 68 |
| Davis Strait | | 149 | 157 | 247 | 90 |
| North Atlantic | | 143 | 145 | 234 | 89 |
| Greenland Sea | | 163 | 166 | 231 | 65 |
| Lincoln Sea | | 166 | 167 | 230 | 63 |






**Table 2.** Trends from 1979 to 2015 in length of open water season, together with trends in
melt onset, freeze-up and melt season duration for 5 sea ice regions (excluding the North
Atlantic where little sea ice exists). Corresponding trends in melt onset, freeze-up and
duration are also shown for the Greenland drainage basins. Only values for the continuous
melt onset and freeze-up periods are listed. Trends are given as days per decade. Statistical
significance of trend at 95 and 99% are denoted by $^+$ and $^{++}$, respectively.

| Region | Open Water Trend (days/dec) | EMO Trend (days/dec) | MO Trend (days/dec) | FO Trend (days/dec) | Melt Duration Trend (days/dec) |
|---|---|---|---|---|---|
| *Sea Ice Regions* | | | | | |
| Baffin Bay | 12.6$^+$ | -5.7$^{++}$ | -8.3$^{++}$ | 7.8$^{++}$ | 16.1$^{++}$ |
| Davis Strait | 15.9$^+$ | -4.7$^+$ | -6.7$^{++}$ | 5.0$^{++}$ | 11.7$^{++}$ |
| North Atlantic | N/A | -6.9$^{++}$ | -7.3$^{++}$ | 8.9$^{++}$ | 16.3$^{++}$ |
| Greenland Sea | 15.2$^+$ | -6.7$^{++}$ | -3.8$^+$ | 2.1 | 5.9$^+$ |
| Lincoln Sea | -0.1 | -4.0$^{++}$ | -3.9$^{++}$ | 1.6 | 5.5$^{++}$ |
| *Greenland Ice Sheet Drainage Regions* | | | | | |
| Baffin Bay | | -6.1$^{++}$ | -6.4$^{++}$ | 4.6$^+$ | 11.1$^{++}$ |
| Davis Strait | | -6.3$^{++}$ | -10.5$^{++}$ | 8.2$^{++}$ | 18.7$^{++}$ |
| North Atlantic | | -10.7$^{++}$ | -16.4$^{++}$ | 5.7$^{++}$ | 22.1$^{++}$ |
| Greenland Sea | | -6.1$^{++}$ | -6.8$^{++}$ | 7.6$^{++}$ | 14.4$^{++}$ |
| Lincoln Sea | | -5.1$^{++}$ | -5.9$^{++}$ | 6.8$^{++}$ | 12.7$^{++}$ |




**Table 3**. Percentage of grid cells with a significant correlation at α = 0.05 relative to the
total grid cells of the unmasked ice sheet. The correlation is between ice sheet meltwater
production and area-averaged sea ice concentration anomalies in the Beaufort Sea and
Baffin Bay (hatched regions in Figures 5a and 6a, respectively.

| Month | Beaufort Sea | | Baffin Bay | |
|---|---|---|---|---|
| | Simple Correlation (%) | Partial Correlation (%) | Simple Correlation (%) | Partial Correlation (%) |
| June | 87.0 | 81.0 | 17.3 | 20.2 |
| July | 31.2 | 13.4 | 11.1 | 2.1 |
| August | 32.6 | 12.5 | 12.5 | 9.4 |





Table 4. Summary table of results discussed in the main body of the manuscript

| Analysis Performed | Davis Strait | Baffin Bay | Lincoln Sea | Greenland Sea | North Atlantic |
|---|---|---|---|---|---|
| SVD: GrIS <> SIC (Fig. 3) | | June | | | |
| SIC trends (Fig. 4) | Reduced in all seasons | Reduced in all seasons | No change | Positive near the coast in spring and winter | N/A |
| Open water days (Fig. 4) | Increase | Increase | Increase | Increase | Increase |
| OWF trends (Fig. 5) | Positive throughout shoulder seasons | Sharp peak in June and October | mixed | Positive throughout year, no sharp peaks | N/A |
| Relative start of melt on SIC and GrIS (Table 2) | SIC MO earlier, SIC FO later | SIC MO earlier, SIC FO later | SIC and GrIS similar | SIC MO earlier, SIC FO later | N/A |
| Trends in timing of EMO, MO, FO (Table 2, Fig. 6) | MO earlier FO later | MO earlier FO later | MO earlier FO later | MO earlier FO later | N/A |
| Synchronicity between GrIS and SIC EMO,MO,FO time series | | R > 0.6 for MO, FO R > 0.5 for detrended data | R > 0.6 for MO, FO R > 0.5 for detrended data | R > 0.5 for EMO | |
| Latent heat fluxes (Fig. 7) | positive all year | positive all year | positive in summer | positive all year | N/A |
| Sensible heat fluxes (Fig. 7( | Positive spring/fall | Positive JASO | Negative all year | Positive spring/fall | N/A |
| Early/late MO years composites (Fig. 7) | Positive in winter, negative rest of year | Majority of positive anomalies | mixed | mixed | |
| Net longwave fluxes (Fig. 8) | Positive anomalies in spring | Positive anomalies in spring | Positive anomalies in spring | Positive anomalies in spring | N/A |





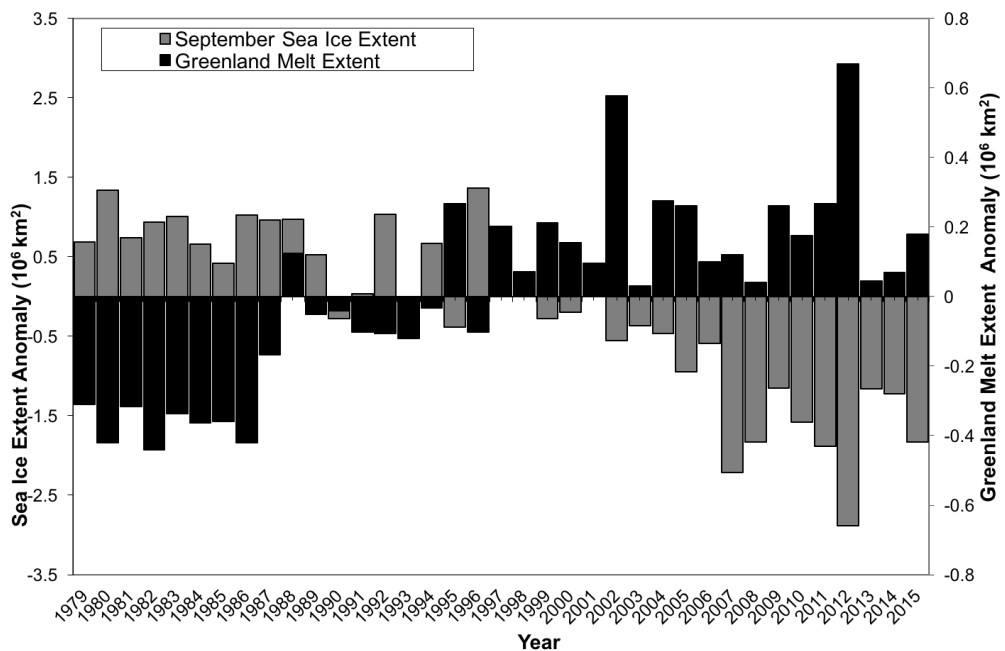

**Figure 1.** Time-series of September sea ice extent and Greenland surface melt extent
anomalies from 1979 to 2015.

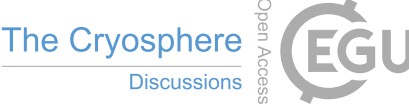




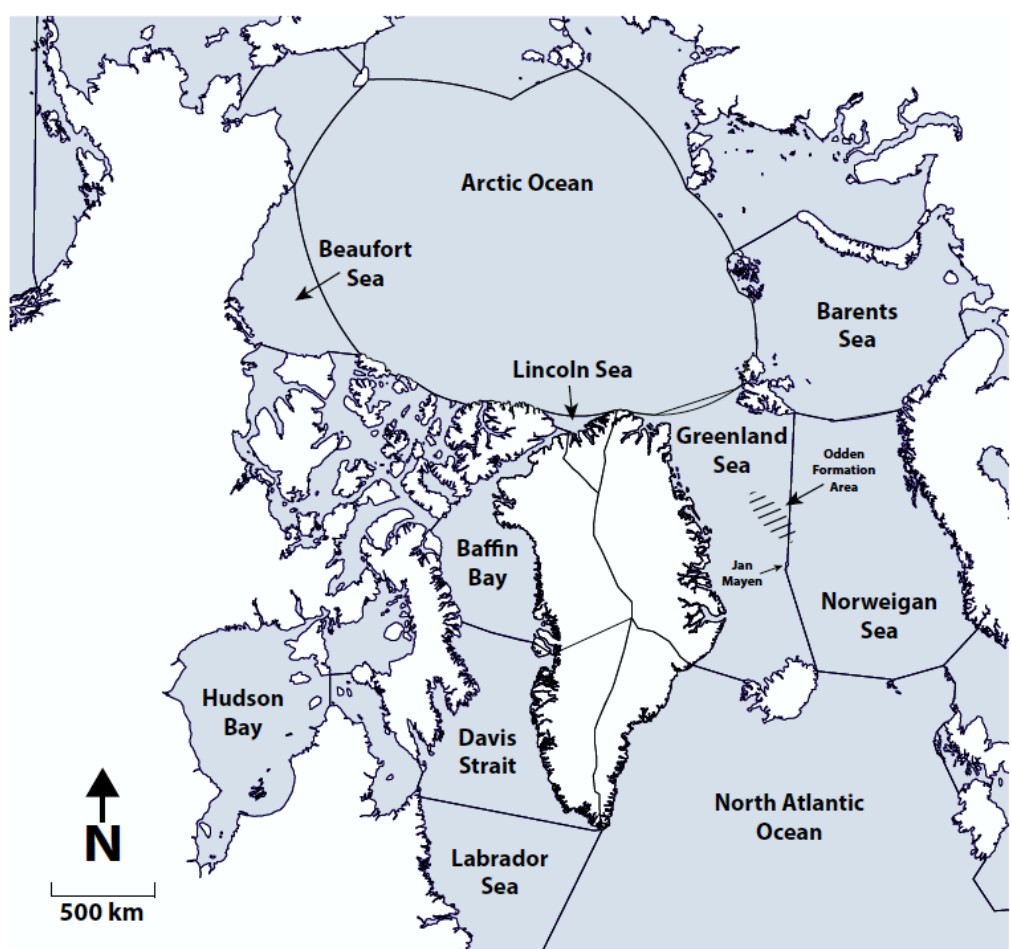

**Figure 2**. Map of study area, including the six sea ice and Greenland drainage sectors
used in this study. The ice sheet regions are named after their adjacent sea (i.e. Davis
Strait, Baffin Bay, Lincoln Sea, Greenland Sea, and the North Atlantic). The
approximate area where the Odden sea ice featured used to formed is indicated with
hatched lines. The ocean boundaries are defined by the International Hydrographic
Organization (VLIZ (2005). IHO Sea Areas. Available online at
http://www.marineregions.org/.)





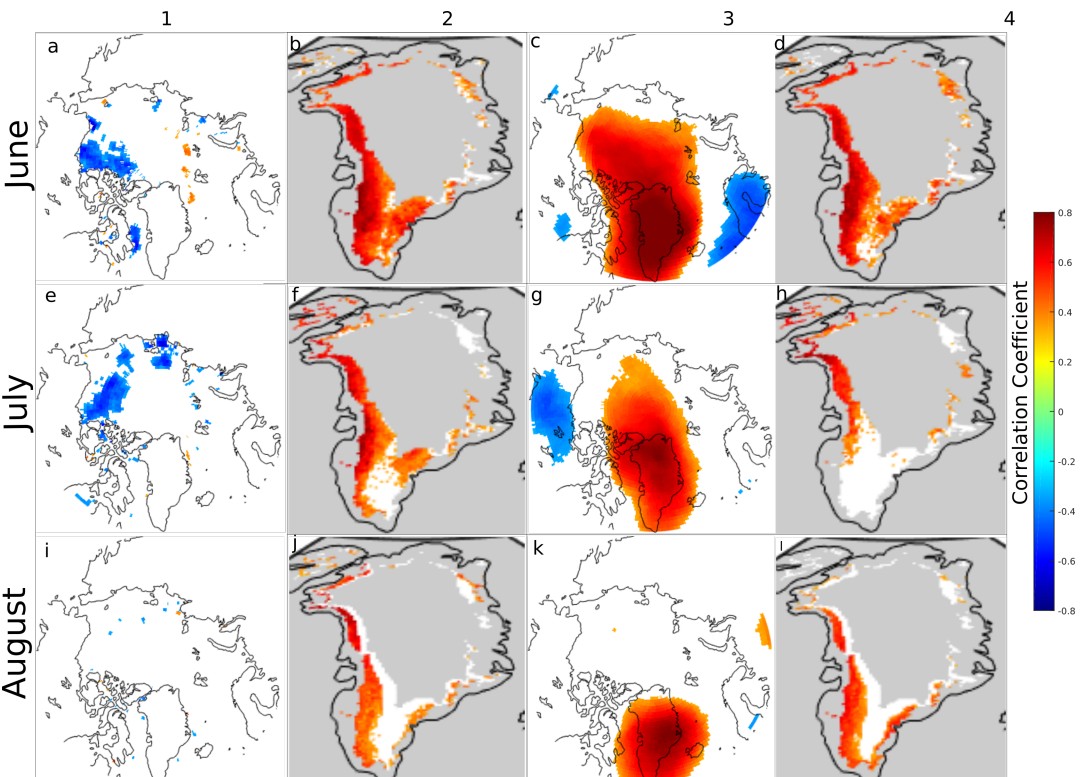

**Figure 3.** Heterogeneous correlation between variables in the leading SVD mode in
JJA. Column 1 is the correlation between sea ice concentration and $EC_{GrIS}$. Column 2 is
the correlation between meltwater production and $EC_{SIC}$. Column 3 is the correlation
between 500 hPa geopotential heights and $EC_{GrIS}$. Column 4 is the correlation between
meltwater production and the $EC_{500}$. Correlation coefficients are not considered over
the masked gray regions, and only correlations significant at $\alpha = 0.05$ are shown. All
data are anomalies relative to 1979-2015 means with the least-squares trend line
removed.



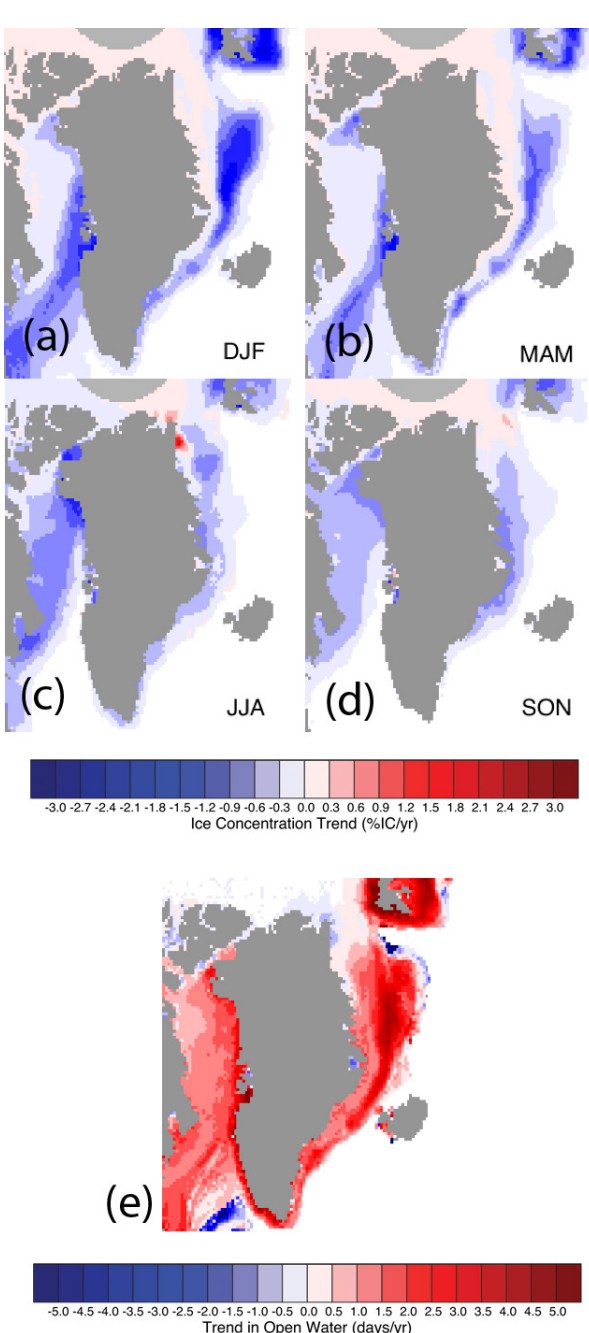

**Figure 4.** Seasonal trends in sea ice concentration from 1979 to 2015 (a-d) and
number of ice free days (e).





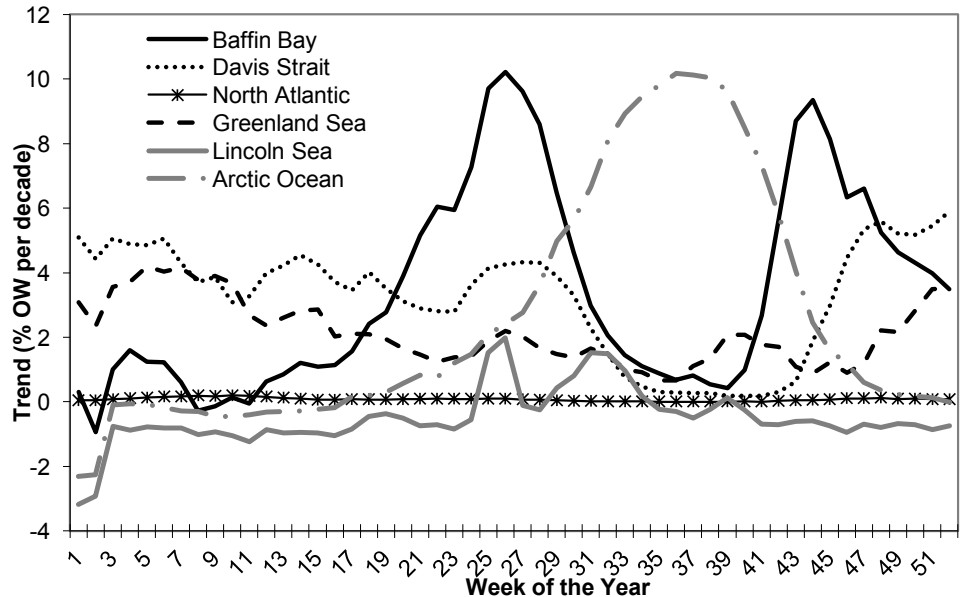

**Figure 5**. Trends in regional open water fraction (OWF) surrounding the Greenland
Ice Sheet, computed from 1979 to 2014.





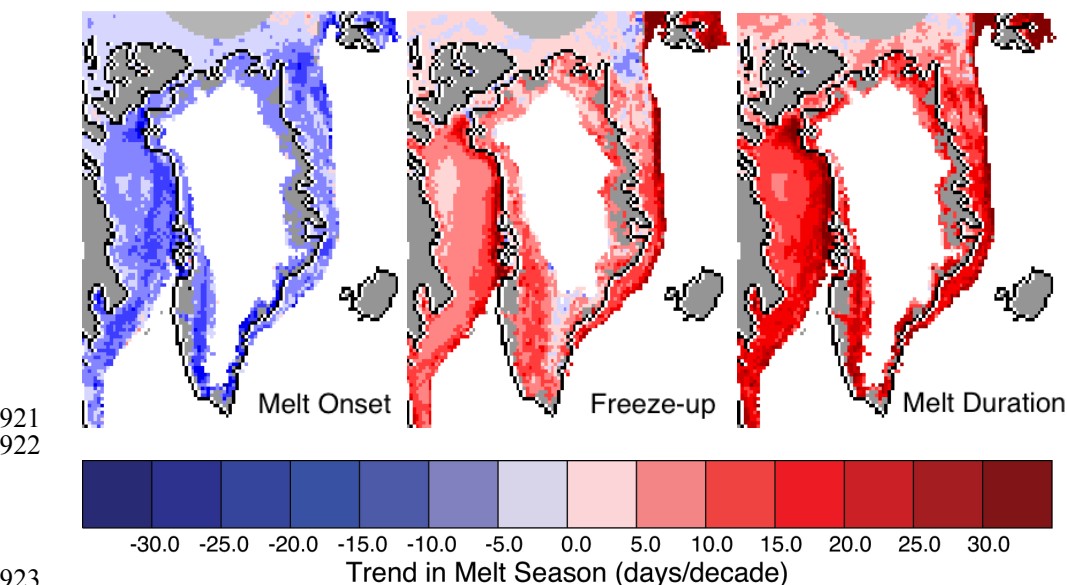



**Figure 6.** Trends in melt onset (top), freeze-up (middle) and total melt season length
(bottom) for sea ice and Greenland from 1979 to 2015.






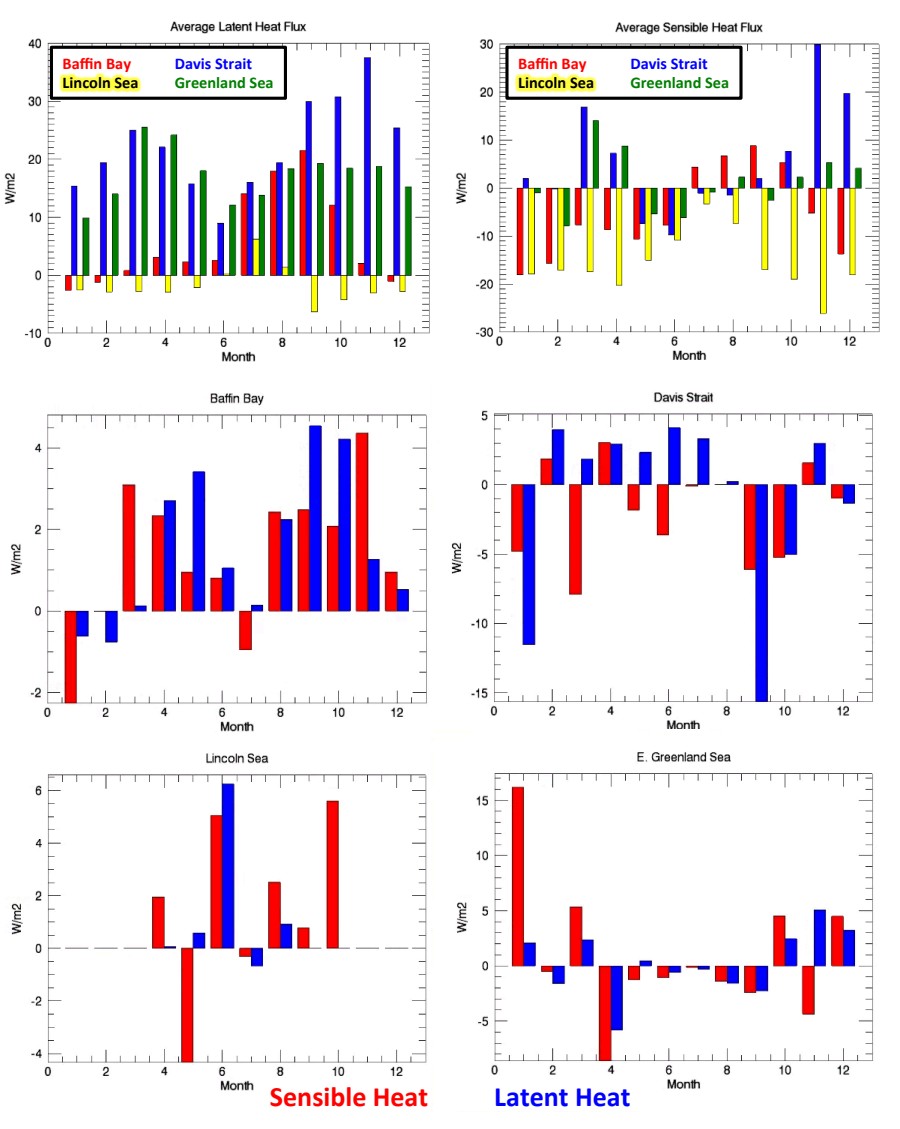

**Figure 7.** Top row graphs show the 2002 to 2015 average latent and sensible heat
fluxes for each ocean region (denoted by color). The sign convention is such that
positive fluxes are directed from the ocean to the atmosphere. Bottom two row



graphs show the early minus late melt onset years for each region of the positive (into
the atmosphere) sensible (red) and latent (blue) heat fluxes.

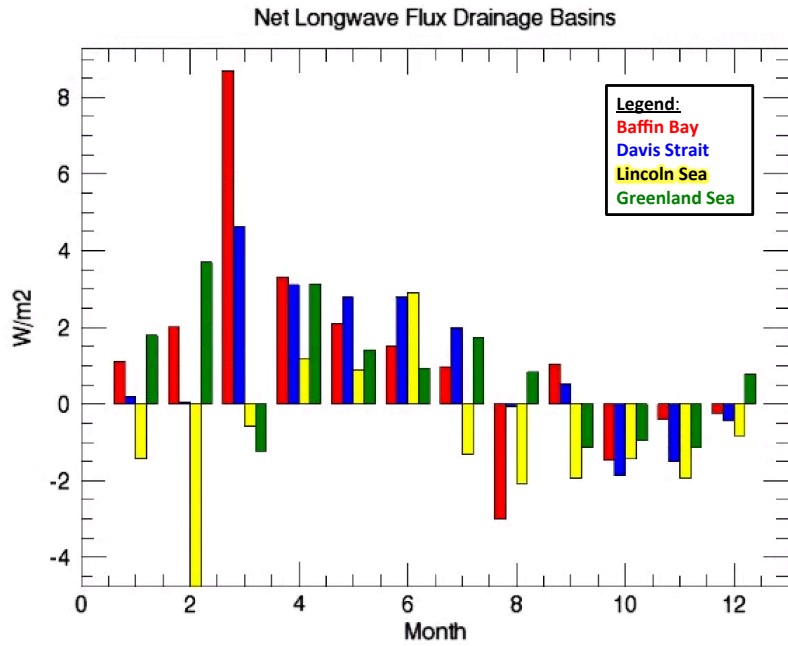

**Figure 8.** Net longwave flux (downwelling longwave flux – upwelling longwave flux)
for early MO minus late MO years for the drainage basins of the Greenland Ice Sheet,
where red bars are for Baffin Bay, blue bars are for Davis Strait, yellow bars are for
Lincoln Sea and green bars and for Greenland sea.






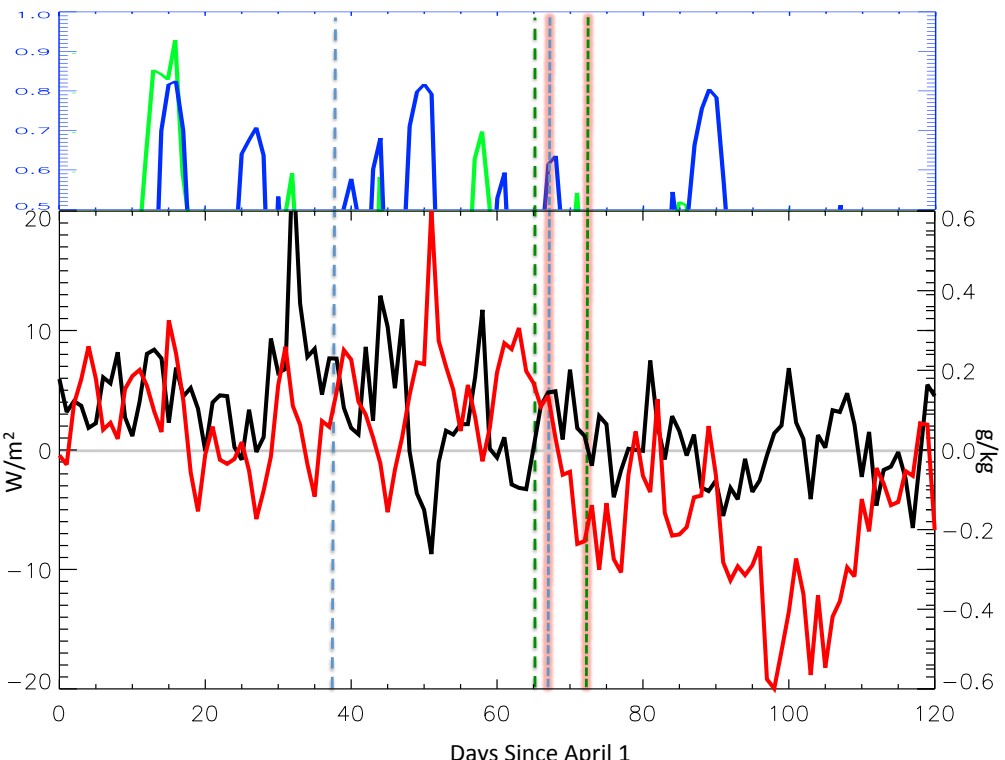

**Figure 9a**. Baffin Bay SIC region latent heat flux from early minus late MO years
(black line) and Baffin Bay GrIS region specific humidity from early minus late MO
years (red line). Dotted vertical lines represent the average early melt onset date for
Baffin Bay (dotted blue), and average late melt onset date for Baffin Bay (dotted blue,
red highlight), average early melt onset date for GrIS (dotted green), and average late
melt onset date for GrIS (dotted green, orange highlight). The top portion of this
figure shows the week lag-1 week lagged running correlations (between 0.5 and 1.0)
for early melt years latent heat flux from Baffin Bay and specific humidity from GrIS
(blue) and late melt onset latent heat flux from Baffin Bay and specific humidity from
GrIS years (green).




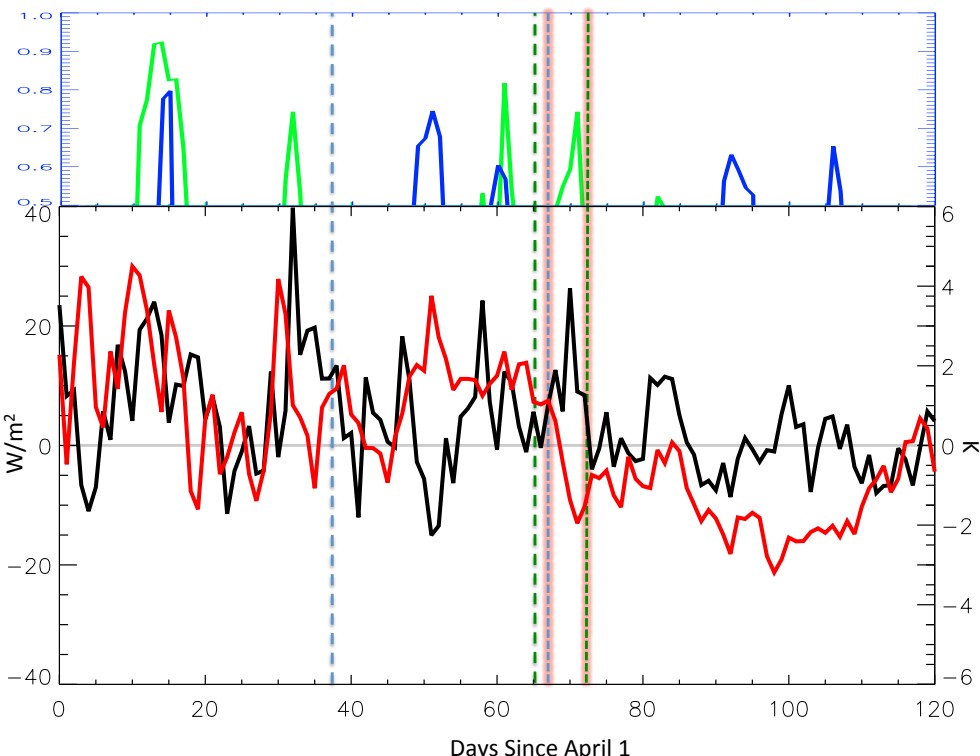

**Figure 9b**. Baffin Bay SIC region sensible heat flux from early minus late MO years
(black line) and Baffin Bay GrIS region air temperature from early minus late MO
years (red line). Dotted vertical lines represent the average early melt onset date for
Baffin Bay (dotted blue), and average late melt onset date for Baffin Bay (dotted
green), average early melt onset date for GrIS (dotted blue, red highlight), and
average late melt onset date for GrIS (dotted green, orange highlight). The top portion
of this figure shows the week lag-1 week lagged running correlations (between 0.5
and 1.0) for early years sensible heat flux from Baffin Bay and air temperature from
GrIS (blue) and late melt onset sensible heat flux from Baffin Bay and air temperature
from GrIS years (green).






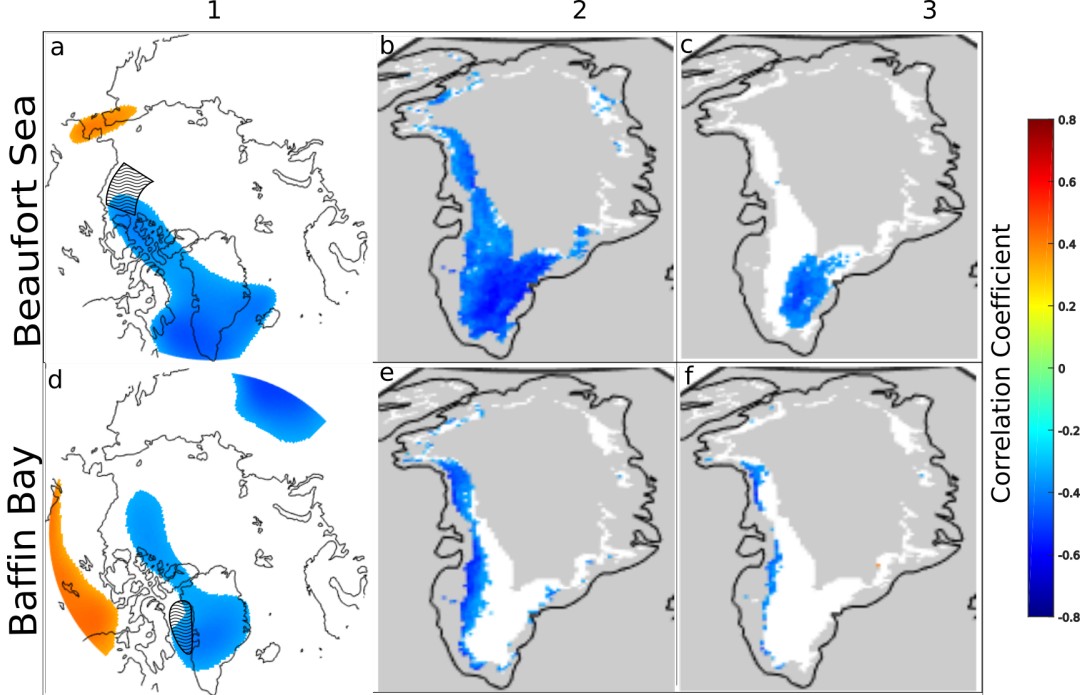

**Figure 10.** June correlation between spatially averaged SIC in the hatched region and: Column 1) 500 hPa geopotential height field, Column 2) Greenland meltwater production, and Column 3) same as Column 2 but with the effect of the Greenland Blocking Index removed (partial correlation). Correlation coefficients are not considered over the masked gray regions. All data are anomalies relative to 1979-2015 means with the least-squares trend line removed.

985
986





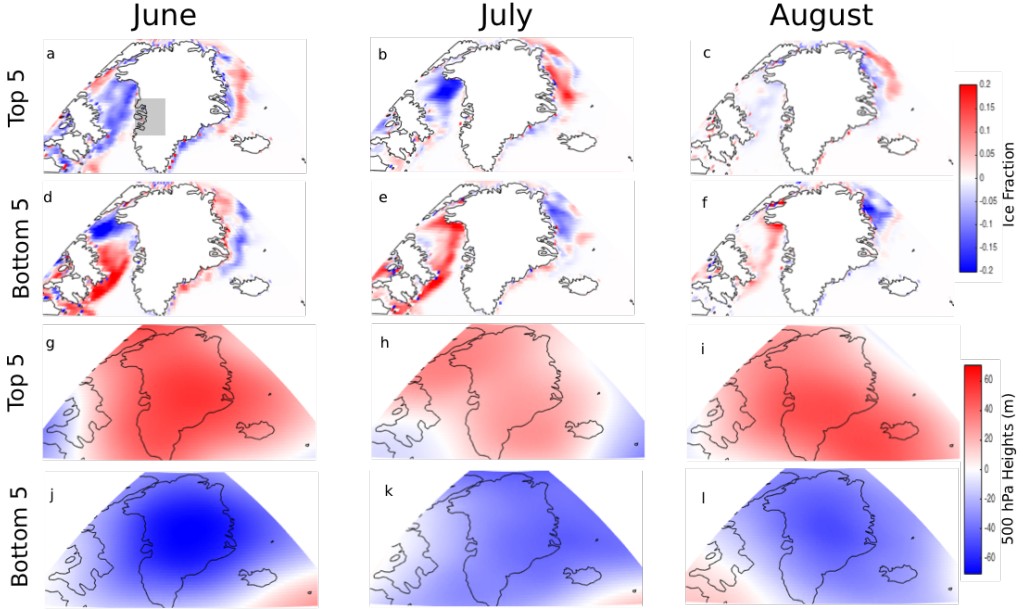

**Figure 11.** De-trended anomalies of SIC (a-f) and 500 hPa geopotential heights (g-l) averaged over the 5 highest and lowest melt years in June, July, and August as indicated by de-trended meltwater production anomalies in the indicated gray region of the ice sheet. Units are ice fraction (a-f) and m (g-l).






Late Melt Years


**Figure 12.** Daily wind vectors at 10 meters from AIRS during 3 early sea melt years
over Baffin Bay (top panel) and 4 late sea melt years (bottom) panel.