# Peer review of "Investigating the Local Scale Influence of Sea Ice on 2 Greenland Surface Melt"

_The Cryosphere, 2017_

## Short Comment (SC1) · 20 May 2017

To the Authors,

First, I was excited to read this Discussion paper, and start to decipher the main results. This is mostly since I'm part of a group exploring (using GCM output) the impact of climate variability on regional Antarctic snowfall. Part of this work involves exploring the impact of regional sea ice variability on regional AIS snowfall (or lack thereof). So there are strong analogies between the two studies.

Secondly, a minor point; I'd noticed you'd cited Fyke et al. (2014) a few times. There are actually two Fyke et al. (2014) papers that I think you may be convolving into a single citation. First one (which you cite) describes an increase in variability in integrated SMB due to, primarily, increased ablation area. Second one

(http://onlinelibrary.wiley.com/doi/10.1002/2014GL060735/abstract) involves the detection of emergence of an anthropogenic signal in GrIS SMB (which you seem to describe, but wrongly cite as results from the 'variability' paper). Perhaps in the final revisions this could be clarified, one way or another.

Again, a minor technical issue, which is definitely overshadowed by the neat aspects of this climate dynamics study.

Thanks

Jeremy Fyke

―――――――――――――――――――

---

## Author Comment (AC1) · 21 May 2017

Hi Jeremy thank you for pointing out our mistake with your references, we will be sure to correct that in the revisions.

---

## Referee Comment (RC1) · Anonymous Referee #1 · 5 Jun 2017

June 5, 2017

I have read the manuscript "Investigating the local influence of sea ice on Greenland surface melt" by J.C. Stroeve et al. This work evaluates statistical and physical links between Arctic sea ice conditions and subsequent melt events observed over the Greenland Ice Sheet. Through a statistical framework, the authors find strong covariability between Baffin Bay and Davis Strait melt and freeze onset and ice sheet melt occurrence within close spatial proximity to the aforementioned oceanic regions. The physical associations presented between local sea ice cover and the ice sheet appear to substantiate the author's statistical findings. In particular, composite turbulent flux and wind field analyses show transport of warm, moist air from the ocean (evident in early melt years) onto the ice sheet that subsequently enhance glacial melt, especially

at lower elevations around the western and southern margins.

Overall, this manuscript is well-written and concise and I believe the quality of analyses presented are consistent with manuscripts published in The Cryosphere. This paper would make a contribution to the growing body of local/regional sea ice-ice sheet interactions within a rapidly changing North Atlantic Arctic environment. I would recommend acceptance pending the completion of a relatively small number of revisions, which I have detailed below by line numbers in the submitted manuscript.

Minor Comments:

Line 56: Add "...and mid-tropospheric height" after SLP to reflect coincident mid-level circulation changes in the Arctic. Papers such as Bezeau et al (2015) Int J Clim (doi:10.1002/joc.4000) could also be cited here.

Line 60: Relevant recent work by Ballinger et al. (2017) Clim Dyn (doi:10.1007/s00382-017-3583-3) similarly notes poleward advection of warm air masses delays autumn freeze onset in Baffin Bay, and impacts Greenland coastal temperature signals, and would appropriately fit here.

Line 129: Do MAR 850 hPa winds, which use ERA products, compare more favorably relative to observations than MERRA low-level winds? As 500 hPa geopotential height and 10m winds are obtained from MERRA it would seem appropriate to use a similar product for 850 hPa winds.

Lines 132-133: Clarify whether MAR output for 2002 is forced by ERA-40 and ERA-I or just one of these datasets.

Line 158: List the threshold of statistical significance used throughout the paper.

Line 193: Change "didn't" to "did not"

Lines 195-198: The authors should explicitly state what advantages SVD offers beyond more traditional bivariate correlation between two data fields? Such justification would

be helpful given that simple and partial correlation techniques are also utilized in the paper.

Line 239: Should this be Eq. 1? I do not see a second equation listed in the manuscript.

Line 241-242: When compositing by anomalous melt and freeze years using a +/-1 sigma threshold, it appears that only 3 early melt and 4 late melt onset are considered (as mentioned in Fig 12). If the sigma threshold is relaxed to increase sample size (perhaps to +/-0.75 sigma) does this substantially alter lower tropospheric wind patterns?

Line 280: Change "hPA" to "hPa."

Table 3: Does simple correlation reference a specific technique (i.e. Pearson's or Spearman's)? Clarify this in the caption and table.

Figures 9a/b: Graphic is somewhat confusing with time series plots stacked directly on top of each other. I would suggest that panels be clearly separated into a two-panel plot (labeled as a-d for instance) with y axis labeled accordingly on the correlation time series.

Figure 12: Are these winds from AIRS or MERRA? The manuscript explicitly mentions use of MERRA 10m winds (line 160), but not from AIRS.

―――――――――――――――――――

---

## Referee Comment (RC2) · Anonymous Referee #2 · 4 Jul 2017

**General comments**

The study documents a statistical investigation and possible dynamical explanations of the covariance between sea ice concentration and the surface melt of Greenland Ice Sheet. The purpose is to demonstrate the impact of local changes of sea ice around Greenland on the ice sheet surface mass balance. The manuscript is well structured and the conclusions are clear based on the results of the analyses. The topic is interesting and relevant within the scope of The Cryosphere, and thus, with the changes suggested below, it should be acceptable for publication.

**Specific comments**

L254: you stated that the leading SVD mode explains 62% covariance between SIC

and GrIS melt water production in June. This number might be misleading if there is only a very small amount of covariance between these two fields. In this case, the 'normalized squared covariance' (NSC) should be included as well (see details in Wallace et al., 1993 Journal of Climate).

L262: Figure 3(i) does not show any significant HC values, so it should not be included in this sentence.

L482-492: after you remove the impact of Greenland Blocking, the Baffin Bay SIC influence becomes much smaller. This may be because the overall coupling between the SIC and GrIS surface melt is not significant. Thus, the additional calculation of NSC mentioned above will probably help you to interpret the results.

L543-552: you wanted to show that during the early MO years over the sic ice, the wind is blowing from the open water areas onto the GrIS, but the plots in Figure 12 are exactly the opposite. For example the figures in upper panel, the winds are offshore along the west coast of Greenland in all three cases.

Figure4: can you please tell more details of how you calculated the linear trends? Which method you used to do the significant test? In the legend, you should also specify which confidence level you used.

Figure5: maybe you can indicate the weeks with significant trends on different lines?

Figure6: please specify the confidence level you used. Maybe use 'a, b and c' to mark the figures in order to be consistent with other figures? Or, it should be 'left, middle and right', but not 'top, middle and bottom'.

Figure 12: please specify what is the parameter shown with blue and red colors. I guess it's SIC, and if so, you should give a separated color bar as well. Please make the color bars and the wind vectors bigger and clearer. The readers can hardly see them.

[Figure]

---

## Author Response (AR1)

June 5, 2017

I have read the manuscript "Investigating the local influence of sea ice on Greenland surface melt" by J.C. Stroeve et al. This work evaluates statistical and physical links between Arctic sea ice conditions and subsequent melt events observed over the Greenland Ice Sheet. Through a statistical framework, the authors find strong covariability between Baffin Bay and Davis Strait melt and freeze onset and ice sheet melt occurrence within close spatial proximity to the aforementioned oceanic regions. The physical associations presented between local sea ice cover and the ice sheet appear to substantiate the author's statistical findings. In particular, composite turbulent flux and wind field analyses show transport of warm, moist air from the ocean (evident in early melt years) onto the ice sheet that subsequently enhance glacial melt, especially at lower elevations around the western and southern margins.

Overall, this manuscript is well-written and concise and I believe the quality of analyses presented are consistent with manuscripts published in The Cryosphere. This paper would make a contribution to the growing body of local/regional sea ice-ice sheet inter- actions within a rapidly changing North Atlantic Arctic environment. I would recommend acceptance pending the completion of a relatively small number of revisions, which I have detailed below by line numbers in the submitted manuscript.

Minor Comments:

Line 56: Add ". . .and mid-tropospheric height" after SLP to reflect coincident mid-level
circulation changes in the Arctic. Papers such as Bezeau et al (2015) Int J Clim
(doi:10.1002/joc.4000) could also be cited here.
Thank you for bringing this new reference to our attention. We have made the suggested
additions.
Line 60: Relevant recent work by Ballinger et al. (2017) Clim Dyn (doi:10.1007/s00382-
017-3583-3) similarly notes poleward advection of warm air masses delays autumn freeze
onset in Baffin Bay, and impacts Greenland coastal temperature signals, and would
appropriately fit here.
Thank you for bringing this new reference to our attention. We have added the citation.

Line 129: Do MAR 850 hPa winds, which use ERA products, compare more favorably relative to observations than MERRA low-level winds? As 500 hPa geopotential height and 10m winds are obtained from MERRA it would seem appropriate to use a similar product for 850 hPa winds.
We had at one point used 10 m and 850 hPa winds in Fig. 11 but did not keep them in the end. It looks like we forgot to remove that from the data section so this becomes a non-issue. However, we used 10 m wind in Fig. 12 from MERRA-2.

Lines 132-133: Clarify whether MAR output for 2002 is forced by ERA-40 and ERA-I or just one of these datasets.
It is forced by ERA-40 from 1979-2002 and by ERA-I from 2002 to 2015.

Line 158: List the threshold of statistical significance used throughout the paper.
Done

Line 193: Change "didn't" to "did not"
Done

Lines 195-198: The authors should explicitly state what advantages SVD offers beyond more traditional bivariate correlation between two data fields? Such justification would be helpful given that simple and partial correlation techniques are also utilized in the paper.
The advantage of singular value decomposition is that it is able to maximum the covariance between the two fields to explicitly show the structure of the covariability, and also provides subsequent orthogonal modes of covariability (not as relevant for this paper since we only show the leading mode) that are by definition unrelated to the leading mode.

Line 239: Should this be Eq. 1? I do not see a second equation listed in the manuscript.
Thank you, this was a typo now corrected.

Line 241-242: When compositing by anomalous melt and freeze years using a +/-1 sigma threshold, it appears that only 3 early melt and 4 late melt onset are considered (as mentioned in Fig 12). If the sigma threshold is relaxed to increase sample size (perhaps to +/-0.75 sigma) does this substantially alter lower tropospheric wind patterns?
The 1σ threshold gives a different number of years for each region. In Figure 12 we just show the Baffin Bay region which does have only a few years with 1s differences in MO. If we relax to 0.75σ, the number of years anomalous in the AIRS time-period did not change. Ideally, we need a longer time-series of data from AIRS to look at including even more anomalous years in the composite.

Line 280: Change "hPA" to "hPa."
Done

Table 3: Does simple correlation reference a specific technique (i.e. Pearson's or Spearman's)? Clarify this in the caption and table.
This was a Pearson's correlation, this has now been added to the table caption.

Figures 9a/b: Graphic is somewhat confusing with time series plots stacked directly on top of each other. I would suggest that panels be clearly separated into a two-panel plot (labeled as a-d for instance) with y axis labeled accordingly on the correlation time series.

We have changed the figures accordingly to make them Figures A-D. Here are the
captions for each:
Figure 9.
a) Baffin Bay SIC region latent heat flux (W/m2) from early minus late MO years
(black line) and Baffin Bay GrIS region specific humidity (g/kg) from early minus late
MO years (red line).
b) Week Lag-1week lagged running correlations (between 0.5 and 1.0) for early MO
years latent heat flux from Baffin Bay and specific humidity from GrIS (blue line) and
late MO latent heat flux from Baffin Bay and specific humidity GrIS years (green line).
c) Baffin Bay SIC region sensible heat flux (W/m2) from early minus late MO years
(black line) and Baffin Bay GrIS region air temperature (K) from early minus late MO
years (red line).
d) Week Lag-1week lagged running correlations (between 0.5 and 1.0) for early MO
years sensible heat flux from Baffin Bay and air temperature from GrIS (blue line) and
late MO sensible heat flux from Baffin Bay and air temperature GrIS years (green line).
In a-d) Dotted vertical lines represent the aver early MO date for Baffin Bay (dotted
blue), and average late MO date for Baffin Bay (dotted blue, red highlight), average
early MO date from GrIS (dotted green), and average late MO date form GrIS (dotted
green, orange highlight).

Figure 12: Are these winds from AIRS or MERRA? The manuscript explicitly mentions
use of MERRA 10m winds (line 160), but not from AIRS.
Correct, these winds are from MERRA2 in Figure 12. The reason why we do not use
AIRS in this figure is because AIRS does not produce a wind product and there are no
other satellite based wind products in the Arctic to our knowledge.
New Caption: Figure 12. Wind vectors and speeds at 10 meters from MERRA2 during
early melt years over Baffin Bay (top panel) and 3 late sea ice melt years (bottom
panel). Smaller figures superimposed on the wind maps show the sea ice
concentration (%) for that day.
The study documents a statistical investigation and possible dynamical explanations of the covariance between sea ice concentration and the surface melt of Greenland Ice Sheet. The purpose is to demonstrate the impact of local changes of sea ice around Greenland on the ice sheet surface mass balance. The manuscript is well structured and the conclusions are clear based on the results of the analyses. The topic is interesting and relevant within the scope of The Cryosphere, and thus, with the changes suggested below, it should be acceptable for publication.

**Specific comments**

L254: you stated that the leading SVD mode explains 62% covariance between SIC C1 TCD and GrIS melt water production in June. This number might be misleading if there is only a very small amount of covariance between these two fields. In this case, the 'normalized squared covariance' (NSC) should be included as well (see details in Wallace et al., 1993 Journal of Climate).

The SVD was the first step for that particular analysis because it showed us the actual regions where we could be seeing a causal relationship based the locations of (maybe somewhat confusingly) significant correlations in the heterogeneous correlation maps. From there, we isolated two regions, Baffin Bay and Beaufort Sea, and did a similar analysis using partial correlation. Correlation is limited because you can't have two fields, so I took the spatial average of Baffin and Beaufort and correlated (and partial-correlated) that time series with the time series of melt at each grid point in Greenland.

Thank you for pointing out these references to the NSC. We added reference in the methods to the NSC: The normalized squared covariance (NSC) associated with each pair of spatial patterns indicates the total strength of this relationship [*Wallace et al.* 1993], with values greater than approximately 0.10 considered to indicate a significant relationship [*Riaz et al.* 2017].

We further calculated this normalized squared covariance as suggested. The NSC between 500 hPa heights and melt is:
June: 0.191
July: 0.111
August: 0.093

For sea ice and melt, NSC is:
June: 0.099
July: 0.081
August: 0.066

So there generally is a significant coupling between the ice sheet melt and height fields as we expect, but it is less significant between ice sheet melt and sea ice, which
is more important. We were already thinking that this relationship is rather tenuous,
and this is just further support for that idea. We additionally added the NSC values to
Figure 3.

There is very little information employing NSC, and how hard a threshold of 0.10 is,
but we do believe that the reviewer's comment on 482-492 is justified then and the
fact that we lose much of the correlation between sea ice/ice melt after removing the
GBI warrants mention of this 0.099 value to say that the coupling between these two
fields is relatively weak. We added a statement in the Discussion to highlight this
result:
*This explanation is supported by the relatively weak value of NSC for June GrIS melt and*
*SIC, which nearly doubles to 0.191 in the SVD analysis of 500 hPa geopotential heights*
*instead of SIC.*

L262: Figure 3(i) does not show any significant HC values, so it should not be included in
this sentence.
That is correct and it is a relic of the old figures, so we removed reference to Fig. 3i.

L482-492: after you remove the impact of Greenland Blocking, the Baffin Bay SIC
influence becomes much smaller. This may be because the overall coupling between the
SIC and GrIS surface melt is not significant. Thus, the additional calculation of NSC
mentioned above will probably help you to interpret the results.
See our response to previous comment.

L543-552: you wanted to show that during the early MO years over the sic ice, the wind is
blowing from the open water areas onto the GrIS, but the plots in Figure 12 are exactly the
opposite. For example, the figures in upper panel, the winds are offshore along the west
coast of Greenland in all three cases.
That was a typo on our part, this has now been corrected. The winds do indeed show
onshore flow along the west coast of Greenland during early MO years. The figures have
been made larger so that this is seen easier.

Figure4: can you please tell more details of how you calculated the linear trends? Which
method you used to do the significant test? In the legend, you should also specify which
confidence level you used.
Done, linear trends were computed using least square regression and evaluated using a
student T-test at 95% confidence.

Figure5: maybe you can indicate the weeks with significant trends on different lines?
We feel this will make the figure too cluttered as it is a busy figure already. The figure is
more for illustration, showing how trends vary as a function of time of year for the
different sectors around Greenland.

Figure6: please specify the confidence level you used. Maybe use 'a, b and c' to mark the
figures in order to be consistent with other figures? Or, it should be 'left, middle and right',
but not 'top, middle and bottom'.
Yes thank you, this has been corrected.

Figure 12: please specify what is the parameter shown with blue and red colors. I guess it's
SIC, and if so, you should give a separated color bar as well. Please make the color bars
and the wind vectors bigger and clearer. The readers can hardly see them.
We made the figures larger and added the SIC color bar.

[revised manuscript text omitted]

June 16, 2003    June 18, 2006

June 15, 2010    June 15, 2013

10m wind 6/16/2007

10m wind 6/16/2015

Late Melt Years

10m wind 6/16/2003

10m wind 6/16/2010

[Figure]

**Figure 12.** Wind vectors and speeds at 10 meters from MERRA-2 during 4 early sea melt years over Baffin Bay (top panel) and 3 late sea melt years (bottom) panel. Smaller figures superimposed on the wind maps show the sea ice concentration (%) for that day.